# Role of Facets and Morphologies of Different Bismuth-Based Materials for CO_2_ Reduction to Fuels

**DOI:** 10.3390/ma17133077

**Published:** 2024-06-22

**Authors:** Smritirekha Talukdar, Tiziano Montini

**Affiliations:** Environment and Transport Giacomo Ciamician, Consortium INSTM, UdR Trieste and ICCOM-CNR Trieste Research Unit, Department of Chemical and Pharmaceutical Sciences, Center for Energy, University of Trieste, Via L. Giorgieri 1, 34127 Trieste, Italy; smritirekha.talukdar@phd.units.it

**Keywords:** CO_2_ reduction, bismuth, facet, morphology, electrocatalysis, photocatalysis, photoelectrocatalysis

## Abstract

Carbon dioxide (CO_2_) emission has been a global concern over the past few decades due to the increase in the demand of energy, a major source of which is fossil fuels. To mitigate the emission issues, as well as to find a solution for the energy needs, an ample load of research has been carried out over the past few years in CO_2_ reduction by catalysis. Bismuth, being an active catalyst both photocatalytically and electrocatalytically, is an interesting material that can be formed into oxides, sulphides, oxyhalides, etc. Numerous works have been published based on bismuth-based materials as active catalysts for the reduction of CO_2_. However, a proper understanding of the behavior of the active facets and the dependence of morphology of the different bismuth-based catalysts is an interesting notion. In this review, various bismuth-based materials will be discussed regarding their activity and charge transfer properties, based on the active facets present in them. With regard to the available literature, a summarization, including photocatalysis, electrocatalysis as well as photoelectrocatalysis, will be detailed, considering various materials with different facets and morphologies. Product selectivity, varying on morphological difference, will also be realized photoelectrochemically.

## 1. Introduction

Humans have long depended on fossil fuels ever since their discovery in the 13th century [1]. Since then, fossil fuels have been used extensively to meet energy needs, which has led to gradual disruption in the ecological balance of the earth [2,3]. As of now, the CO_2_ level in the atmosphere has reached around 425 ppm according to Mauna Loa Observatory (MLO), Hawaii [4]. This gradual increment in carbon dioxide (CO_2_) levels has been as such for the last 200 years, which has ultimately resulted in the warming up of the Earth at 0.12 °C per decade [5]. Mitigation strategies for CO_2_ include several approaches, amongst which CO_2_ capture and storage (CCS) and CO_2_ capture and utilization (CCU) are some of the most important ones [6]. CCS involves the sequestration of gaseous CO_2_ by absorbing materials that can mineralize CO_2_ to carbonates. However, long-term storage safety and stability pose an issue. CCU includes the added advantage of utilizing CO_2_, in addition to strategies for its storage as compared to CCS. The idea involves converting captured CO_2_ into useful products, thereby inculcating a negative carbon footprint while obtaining the necessary value-based chemicals. CO_2_ chemical fixation (CCF), which is a type of strategy in CCU, involves the recycling of CO_2_ into valuable carbon-containing products. The benefits involve the convenient storage of liquid CO_2_-derived products in ambient conditions, the ability to power an overall carbon-neutral energy cycle, and the ability to produce key commodity chemicals from CO_2_ as the C1-building block via its selective reduction to formic acid (HCOOH), methane (CH_4_), carbon monoxide (CO), methanol (CH_3_OH) and/or C-C coupling products [7]. Therefore, to counteract the evident problem of increased CO_2_ levels, as well as to suffice the energy demands, CCF can be a promising and smart approach [8].

CO_2_ is a linear centrosymmetric molecule with a large bonding energy of C=O double bond (~750 kJ mol^−1^) [9]. Thermodynamically, products can be obtained from the reactants only if the system’s free energy is lowered after the reaction. Therefore, in the case of CO_2_ reduction reactions, the electrons involved in that particular reaction must have a chemical potential (i.e., free energy) higher than the free energy required to drive the reaction. This means G (reactants) ≥ G (products), where G denotes Gibb’s free energy. A potential where these two values become equal denotes the reduction potential of the reaction [10].

As can be seen from Table 1, the reduction potential for CO_2_/CO_2_^•−^ is as negative as −1.9 V. Forming this one-electron reduced intermediate (CO_2_^•−^) is an energy-intensive process, along with being thermodynamically unfavorable. Hence, to obtain feasible reactions with low overpotential and to make the reactions energetically favorable, the need for catalysts comes to light. Another important factor to be considered for CO_2_ reduction reactions (eCO_2_RR) is the selectivity of products. Different reaction pathways can yield different products which depend on various factors like pH, the overpotential, and the temperature. Therefore, to obtain the desired product selectivity and efficiency, the reaction pathways need to be understood and developed and, hence, suitable catalysts need to be used. Other than the mentioned factors, product selectivity is seen to be affected by the active facets and the morphology of the catalyst.

In general, CO_2_ can be reduced by photocatalysis, electrocatalysis, or even photo-electrocatalysis. Photocatalytic CO_2_ reduction simply begins with the absorption of photons by the semiconductor photocatalyst possessing energy greater than its band gap. This leads to charge separation, resulting in the formation of electrons and holes. Then, surface reactions are involved, where CO_2_ interacts with photogenerated electrons, thus reducing CO_2_ into other carbon-based products. In the case of electrocatalytic CO_2_ reduction reactions, the reaction takes place in the electrical double layer formed by the contact of the catalyst surface in the cathodic compartment and the CO_2_ saturated electrolyte. Also, an important point to consider is that in electrochemical CO_2_ reduction reactions (eCO_2_RR), the electrodes are under potentiostatic control and therefore possess negative charge on the surface of the electrode. This influences the formation of a region called the Inner Helmholtz Plane (IHP), in which the process of the bond rearrangement of the CO_2_ molecule occurs. This region is a part of the electrical double layer. Therefore, applied potential, the interaction of the electrolyte ions with the electrode surface, reaction-induced concentration gradients, etc., play an important role in influencing these electrocatalytic reactions. In the case of photo-electrocatalysis (PEC), the benefit of both photocatalysis and electrocatalysis can be realized. The principle of PEC follows with the photogenerated charge separation, as is seen in the case of photocatalysis; however, the difference lies in the addition of an external bias, which promotes this separation with a better rate as compared to simple photocatalytic process [12,13,14].

Considering the available literature, a statistical report on the search results obtained through Scopus is provided for some of the most used metals in catalytic CO_2_ reduction reactions in Figure 1. The keywords used for the analytics were “Element name” + ”Photocatalytic/Electrocatalytic/Photoelectrochemical” + “CO_2_” + “reduction”. Copper is the most widely used metal for catalytic CO_2_ reduction reactions. However, it suffers from poor selectivity of the reaction products, and in order to improve that factor, there have been a number of articles and review reports focusing on the aspect of facet and morphology engineering [15,16,17,18,19,20,21,22]. These reports consider studies conducted in both photocatalysis as well as electrocatalysis, and also photo-electrocatalysis. Amongst other metal-based catalysts, bismuth promises to be a suitable catalyst that is active for CO_2_ reduction reactions, in addition to being beneficial when compared with factors of abundance, cost, and toxicity. The comparison of abundance and price is provided in Table 2. Moreover, compounds such as Cu_2_Se, Cu-Pb, and Pb-based MOF have been used for CO_2_ reduction reactions; however, the use of Se and Pb has been reported to be toxic [23,24,25,26,27]. Also, the electrocatalytic selectivity of bismuth towards formate production from eCO_2_RR is seen to be higher than tin-based and copper-based materials [28,29].

### Methodology

Bismuth is an important metal that, in different forms of oxides, oxyhalides, perovskites, etc., doped or formed into a composite, can act as an excellent catalyst [5]. There are some reports on facet-controlled bismuth-based materials for CO_2_ reduction reactions [30,31,32,33,34]. However, a comprehensive study on the role played by facet exposure, and the morphological change in bismuth-based materials in photocatalysis, electrocatalysis, as well as photo-electrocatalysis has not been included yet in the literature. As mentioned before, the selectivity of the desired products and the efficiency of the catalytic reactions can be tuned and controlled by understanding the electron transfer mechanism, the mechanism of activation of the catalyst, synthesis directed towards the exposure of a specific active facet, and specific morphology. This brief review is an overview to grasp and have a better understanding of the role played by the control, tuning of facets, and the morphologies of bismuth-based catalysts. Here, it is focused on understanding the facet-directed charge transfer mechanism, the morphological effect on the selectivity of the products, the morphological changes in a material leading to proper exposure of the active sites, and an overview on the reported literature concerning such catalysts.

**Table 2 materials-17-03077-t002:** Abundance, total mass in Earth’s crust and price of bismuth compared to other elements ^2^.

Element	Abundance (mg kg^−1^), Total Mass in Earth’s Crust	Price ($ kg^−1^), Year
Bismuth	0.009, 2.493 × 10^14^ kg	6.36, 2019
Selenium	0.05, 1.385 × 10^15^ kg	21.4, 2019
Tin	2.3, 6.371 × 10^16^ kg	18.7, 2019
Copper	60, 1.662 × 10^18^ kg	6.00, 2019
Lead	14, 3.878 × 10^17^ kg	2.00, 2019
Gold	0.004, 1.108 × 10^14^ kg	44,800, 2019

^2^ Data obtained from reference [35].

## 2. Selectivity of Products Based on Different Morphologies and Facets Exposed

Product selectivity depends mainly on the reaction kinetics and surface adsorption phenomenon. The reactive adsorption of CO_2_, influenced by the presence of acidic and basic sites in the catalysts along with the different functionalization and oxygen vacancies, influences the formation of the reaction intermediates, which determine the end products [36]. The exposure of these active sites would hugely depend on the morphology of the catalyst. With control over the surface morphology, the catalytic sites, such as oxygen vacancies, stoichiometric defects, and grain edges, could be exposed, whose nature and density depend on the exposed facets [37].

### 2.1. Photocatalysis

The photocatalytic reduction of CO_2_ has been long known, since Fujishima and Honda, in 1972, demonstrated for the first time that solar energy can be stored in the form of chemical bonds by semiconductor photocatalysis [38]. Since then, numerous materials have been synthesized and investigated for the photocatalytic reduction of CO_2_. The synthesis of well-defined crystalline oxides has become of interest, since their intrinsic photocatalytic performances are closely related to their exposed facets, where different facets portray different electronic band structures [39,40].

#### 2.1.1. BiOX

BiOX, or bismuth oxyhalides (X = Cl, Br, I), belong to the family of V-VI-VII ternary oxide materials, which have great performance in behaving as active photocatalysts for CO_2_ reduction. Among the oxyhalides, BiOCl is only active in the UV region, accounting for 4% of the sunlight. On the other hand, BiOBr and BiOI, being active in the visible region, have their limitations due to the wide band gap of BiOBr and the relatively smaller band gap of BiOI. There are numerous reports on the photocatalytic reduction of CO_2_ with BiOX catalysts, where mostly the (001) facet was observed to be active, whether being formed with specific morphologies, being formed into a heterojunction, or being modified with defects, doping, etc. [41,42,43,44,45]. These photocatalytic properties are highly dependent on their layered structures. These layered structures have a non-uniform charge distribution, due to which an internal electric field (IEF) occurs. This non-uniform charge distribution is generated due to the presence of the strong intralayer covalent bonding of [Bi_2_O_2_] slabs and the weak interlayer van der Waals interaction of the halogen double slabs. It then generates the IEF along the crystal orientation, perpendicular to the [Bi_2_O_2_] and halogen layers. This induced IEF contributes to the separation and transfer of the photogenerated charge carriers that enhance the photocatalytic performance. It was reported that {001} facet-dominant BiOCl nanosheets had higher photocatalytic activity as compared to the {010} facet-dominant nanosheets. The direction of the charge separation, as shown in Figure 2a, is facilitated towards [001] direction rather than the [010] direction, since a higher exposure of the {001} facet accounts for the effective charge separation. This was also evident with the enhanced photocurrent and quenched photoluminescence signal of the {001} facet dominant nanosheets as compared to the {010} facet dominant ones [46]. Also, it was found that increasing the exposure percentage of {001} facets of Bi_3_O_4_Cl nanosheets induced a stronger IEF, resulting in higher photocatalytic activity. It is explained in this study that, according to the measured surface voltage and charge density, the IEF magnitude was positively correlated with the percentage of exposed {001} facets (Figure 2b) [46]. Another study by Wang et al. supports the same, where they reported the (001) facet providing more photogenerated charge carriers [47]. However, Li et al., in 2018, worked on how Bi nanoparticles being deposited on the different facets of BiOBr resulted in alternative charge transfer properties. It was found that Bi nanoparticles being deposited on the (010) facet had better surface charge alteration, which is more favorable for interfacial charge separation and transfer as compared to the deposition on (001) facet (Figure 2c). A new charge transfer route was established between the contacted Bi nanoparticles and (010) facets of BiOBr along the path of [Bi_2_O_2_]^2+^ → Bi NPs → Br^−^ [48].

BiOBr nanosheets with dominantly exposed (102) facets were doped with Co and were found to produce CO from photocatalytic CO_2_RR equating to 54.5 µmol g^−1^ [49]_._ In another study, Wang et al., in 2024, studied the introduction of Indium Oxide (IO) on 2-D BiOBr (BOB). A CO production rate of 54.2 µmol g^−1^ for the heterojunction with BOB/IO (B_3_I_1_) was achieved, which was 2.2 times and 11.3 times higher than that of pristine BOB and IO, respectively. A HR-TEM analysis enabled us to determine the lattice spacing of the heterojunction to be 0.277 nm and 0.297 nm, corresponding to the (110) facet of BOB and the (222) facet of IO, respectively [50]. In these cases, along with the exposed facet, the selectivity of the product is also driven by the heterojunction material or the dopant entity.

#### 2.1.2. Bi_2_O_3_

Bi_2_O_3_, with its great oxidizability and appropriate band-gap for photoactivity, can be a great catalyst for the photocatalytic reduction of CO_2_. Shi et al., in 2021, synthesized thin film CuBi_2_O_4_/Bi_2_O_3_ by a spray-pyrolysis calcination method with dominant (020) facets with monoclinic phase. Temperature control in the synthesis played an important role in the (020) facet being the dominant facet, as the synthesis carried out at a temperature below 150 °C did not form Bi_2_O_3_ with an exposed (020) facet. The heterojunction-based catalyst of interest was synthesized at 260 °C. The morphology of this catalyst was such that the nanoparticles of Bi_2_O_3_ were uniformly dispersed on the CuBi_2_O_4_ nanosheets. The TEM image of the (020) faceted catalyst indicated that the interplanar spacings of 0.165 nm and 0.240 nm corresponded to the (332) and (202) facets of tetragonal phase CuBi_2_O_4_, and the interplanar spacings of 0.253, 0.243 and 0.407 nm were assigned to the (031), (130) and (020) facets of the monoclinic phase of Bi_2_O_3_. After visible light irradiation for 12 h in a gas–solid phase catalytic system, the products obtained were CO, CH_4_, and O_2_, with productivities of 247.62, 119.27 and 418.00 µmol/m^2^, respectively, with the (020) facet-directed composite material. It was reported that the exposed (020) facets of Bi_2_O_3_ in the composite had enhanced the performance of the photocatalyst by H_2_O oxidation. This happened due to the improved adsorption property for H_2_O molecules on the (020) facet. However, the stronger hydrophobicity of the composite film surface did not allow for the abundant H_2_O molecules to occupy the adsorbed sites of CO_2_ molecules [51]. In another case, β-Bi_2_O_3_ was formed into a composite with g-C_3_N_4_, where the rod-like morphology of Bi_2_O_3_ was seen [52]. It was interesting to note that under photocatalysis, g-C_3_N_4_ produced 1.8 times less CO as compared to the composite 40% Bi_2_O_3_/g-C_3_N_4_. Additionally, Bi_2_O_3_ did not produce any CO, due to its conduction band (CB) being lower than the reduction potential of CO_2_/CO (−0.52 V vs. NHE). The enhancement in the performance of the composite catalyst can be attributed to the exposed (201) facet of Bi_2_O_3_ in the composite, as evident from the HR-TEM image with 0.318 nm interplanar spacing. It was seen that exposure to the (201) facet resulted in uniform combination and affinity in the 40% Bi_2_O_3_/g-C_3_N_4_ composite [52].

#### 2.1.3. BiVO_4_

BiVO_4_ is an active photocatalyst for the reduction of CO_2_. Das et al. in 2022 improved the photoactivity of BiVO_4_ by forming a heterojunction with WO_3_, where the (110) facet of BiVO_4_ played an important role in the activity of the catalyst. The reaction pathway was checked on the (110) facet of both the pristine and the composite by analyzing the reaction thermodynamics. The formation of composite substantially relaxed the strain in BiVO_4_ that led to a low-resistance fast electron transfer process, benefitting the photocatalytic activity. Also, with extended X-ray absorption fine structure (EXAFS), it was seen that upon relaxation, the Bi–Bi radial distance had elongated due to strain relaxation in the BiVO_4_ lattice. The composites that were developed varied, with different BiVO_4_/WO_3_ ratios of 1:1 (comp11), 1:2 (comp12), and 1:4 (comp14). For the photoreduction of CO_2_, it was found that the conversion of *CHO to the *CH_2_O intermediate was the potential determining step (PDS), where the free energy barrier for PDS on the composite (comp14) was much lesser than on the pristine BiVO_4_ (110), as can be seen on Figure 3. This lessening of free-energy barrier for PDS on comp14, as compared to pristine BiVO_4_ (110), can be attributed to the strain–relaxation in the BiVO_4_ lattice as a result of composite formation with WO_3_. It was observed in this case that CO_2_ was effectively reduced to CH_4_, with a yield of 105 µmol g^−1^ h^−1^ [53]. The high selectivity of BiVO_4_ towards the production of ethanol was observed by Liu et al. in 2009, with a monoclinic, sheet-like morphology [54]. A tetragonal zircon BiVO_4_ with rod-like morphology was also synthesized with the use of PEG; however, the production of ethanol was observed to decrease by a factor of 17 as compared to the monoclinic, scheelite BiVO_4_. It was claimed that ethanol formation favored the monoclinic phase, as CO_3_^2−^ is anchored to the Bi^3+^ sites on the surface through a weak Bi–O bond. This enabled the Bi^3+^ sites to efficiently receive the photogenerated electrons from the V 3d-block bands of BiVO_4_. In this case, the local environment around the Bi^3+^ ion is more strongly asymmetric, which lets the Bi^3+^ ion develop a stronger lone pair character and, hence, a stronger affinity for CO_3_^2−^ [54]. Another study reported a composite based on BiVO_4_ and TiO_2_, which produced methane from a photocatalytic CO_2_ reduction reaction, producing 28 µmol g^−1^ in 8 h as compared to 4 µmol g^−1^ and 12.5 µmol g^−1^ of pure TiO_2_ and BiVO_4_, respectively. The increased production of CH_4_ was attributed to the formation of a heterojunction that improved the charge-separation, improving the photocatalytic reaction [55]. An important work considering the role of facets involved in CO_2_ reduction was reported by Zhou et al. in 2018. A high efficiency in catalyst performance was noticed in the case of BiVO_4_{010}-Au-Cu_2_O as compared to BiVO_4_{110}-Au-Cu_2_O. This behavior was mainly attributed to the formation of a Schottky junction on the reduction facet of the semiconductor, where electron accumulation would occur. Therefore, anchoring a metal on that facet enhanced hot-electron injection into the metal. This is due to the high density of hot electrons on the reduction facet and the enhanced surface electric states on the semiconductor due to metal modification. It ultimately accelerated the electron transfer to the metal. Also, the unidirectional electron transfer route from the semiconductor to the metal resulted in efficient charge separation. CO_2_ photoreduction yielded CH_4_ and CO, with production rates being 2.6 and 1.8 times, respectively, higher for BiVO_4_{010}-Au-Cu_2_O than for BiVO_4_{110}-Au-Cu_2_O [56].

#### 2.1.4. BiOIO_3_

BiOIO_3_ is another bismuth-based photoactive material that is composed of an aurivillius-type (Bi_2_O_2_)^2+^ layer and pyramid layers of polar IO_3_^−^, with an orthorhombic phase structure [57,58]. Chen et al., in 2018, found that control over the layer-growth direction would result in a largely shortened diffusion pathway for the charge carriers in the BiOIO_3_ photocatalyst. In addition, it was also reported that an optimal thickness of the photocatalyst, with an appropriate proportion of the exposed {010} and {100} facets, would render efficient separation for photogenerated electrons and holes on the anisotropic facets. This control over the thickness of the nanoplates yielded a significant increase in the CO evolution rate, 5.42 µmol g^−1^ h^−1^, compared to its bulk counterpart [59]. Again, Chen et al. synthesized BiOIO_3_ single-crystal nanostrips, oriented along the growth direction of [001] with surface oxygen vacancies. Controlling the synthesis parameters, single crystals of BiOIO_3_ with different lengths containing various amounts of IO_3_ polyhedra were formed, where distinct changes in polarity were noticed. The nanostrips with an optimal oxygen vacancy concentration showed a CO production rate of 17.33 µmolg^−1^ h^−1^, which was higher than that of the BiOIO_3_ nanoparticles [60]. Supercritical CO_2_ was used to create the epitaxial heterostructures of Bi_2_O_2_CO_3_/BiOIO_3_. In the study, 2-D facet-coupled heterostructures were synthesized with (110) growth plane of Bi_2_O_2_CO_3_ oriented parallel to the BiOIO_3_ (210) plane. This 2-D facet oriented heterostructure-based composite was seen to exhibit CO_2_ to CO conversion at the rate of 224.71 µmol g^−1^ h^−1^ [61].

#### 2.1.5. Bi_2_S_3_

Bi_2_S_3_ is a binary n-type chalcogenide that is an amiable choice for photocatalyst, due to its well-known narrow band gap, high optical absorption coefficients in the visible light region (>10^5^ cm^−1^) and near infrared range (>10^4^ cm^−1^) [62,63]. Nanostructures of the 1-D morphology of Bi_2_S_3_ crystals are mostly formed due to the restriction of crystal growth along the ab plane once the bonds of Bi^3+^ and S^2−^ are built. Despite being hugely photoactive in the visible and NIR light, the fast recombination of electrons and holes remains a problem for this photocatalyst [63]. Guo et al., in 2020, synthesized Bi_2_S_3_ quantum dots uniformly dispersed over g-C_3_N_4_. The Bi_2_S_3_ quantum dots were spherical in shape and the photocatalytic reduction of CO_2_ using this catalyst yielded CO with a maximum value of 54.74 µmol g^−1^ [64]. In another work, Kim et al., in 2019, reported Bi_2_S_3_ which was shaped in the form of nanorods. A heterojunction with MoS_2_ nanosheets was made, which actively formed CO from CO_2_ after 10 h of irradiation [65]. Bi_2_S_3_ with different shapes, such as nanoparticles and urchin-like hierarchical microspheres, were synthesized by Chen et al. in 2013. The work reported the role of different solvents used in the synthetic system that led to the formation of different morphology of Bi_2_S_3_. The urchin-like morphological structure consisted of numerous nanorods. CO_2_ photocatalytic reduction using these photocatalysts in methanol saturated with CO_2_ yielded formic acid and formaldehyde. These were converted to methyl formate by the esterification of formic acid and methanol and the dimerization of formaldehyde by Tishchenko reaction. It was seen that the yield of methyl formate was more in the case of the urchin-like hierarchical structures as compared to the nanoparticles. One of the main factors, as explained, was the morphology of the material, where the microspheres exhibited multiple reflections in the gaps of the packing nanoplates, as shown in Figure 4b, and, hence, increased the availability of more surface active sites [66].

#### 2.1.6. Bi_2_WO_6_

Bi_2_WO_6_ is composed of accumulated layers of corner-sharing WO_6_ octahedral sheets and bismuth oxide sheets that behave as an excellent photocatalyst [67]. Zhou et al., in 2011, synthesized ultrathin and uniform Bi_2_WO_6_ nanocrystallites, which actively converted CO_2_ into CH_4_ with a yield producing rate of 1.1 µmol g^−1^ h^−1^. The photocatalysis was carried out in a gas–solid system in the absence of any cocatalysts under visible light (λ > 420 nm). These nanocrystallites were in the shape of nanoplates, having ~9.5 nm thickness. This morphological aspect of ultrathin geometry allowed for the fast movement of charge carriers from the interior to the surface, and it improved the electron-hole separation. Tuning the shape of the material also allowed for enhanced photoactivity, as the nanoplate formation led to the exposure of the {001} facet. This is in alignment with the DFT studies, which showed that the adsorption energy for CO+O (0.487 eV) was larger than the adsorption energy of CO_2_ (0.176 eV) on the {001} facet as compared to the {010} and {101} facets. This explains the ease of CO_2_ dissociation on the {001} surface [68]. One work reported on the formation of composites of g-C_3_N_4_ and Bi_2_WO_6_ using a simple hydrothermal method. Nanoflakes of Bi_2_WO_6_ were deposited on the lamellar structure of g-C_3_N_4_ with crystal planes (020) and (113), as confirmed by XRD and SEM images. The composite was mainly able to produce CO, with its highest yield obtained at the 8th hour under visible light irradiation, corresponding to 5.19 µmol g^−1^ h^−1^, which was about 22 and 6.4 times more than g-C_3_N_4_ and Bi_2_WO_6_. It was also evident from the TEM images that the nanoflake morphology of Bi_2_WO_6_ allowed for the formation of actual interfaces with g-C_3_N_4_, rather than just a physical mixture, which ultimately led to an increment in the catalyst performance. This catalyst produced CO with high selectivity but had no particular production of CH_4_, as has been noted with most of the other Bi_2_WO_6_-based catalysts [69]_._ Cheng et al. synthesized hollow microspheres of Bi_2_WO_6_ and obtained methanol as the major product from the photocatalytic reduction of CO_2_. The morphology of the material consisted of hollow microspheres, as was seen with SEM images. Higher magnification provided detailed surface structures, which showed the microsphere being composed of crossed nanosheets [70]. Table 3 provides an overview of the studied literature, comparing the activity of the catalysts discussed above.

Hence, it can be understood that the morphology, as well as the facet exposed, play an important role in photocatalytic CO_2_ reduction reactions. The selectivity of the products are affected greatly even by modulating the morphology of the catalyst.

### 2.2. Electrocatalysis

Electrocatalysis consists of redox reactions that occur on the surface of electrodes. These electrodes have catalysts on their surfaces, which lowers the overpotential of the electrochemical reaction, which in this discussion is a CO_2_ reduction reaction. The interaction of the CO_2_ molecule with the catalyst surface determines a lot of the products to be expected, due to the catalytic reaction. The CO_2_ molecule has 1π_g_ as its highest occupied molecular orbital (HOMO) and 2π_u_ as the lowest occupied molecular orbital (LUMO). These absolute HOMO and LUMO energy levels greatly influence the adsorbate-surface chemistry, as the binding interaction greatly depends on the orientation of these orbitals. These orbitals are distributed symmetrically along the molecular axis, with 1π_g_ involved in donating the electron density to the electrode and the 2π_u_ orbital involved in accepting electrons from the electrode. This results in the formation of new hybrid electronic states. Moreover, chemisorption makes the CO_2_ molecule have a bent configuration [13]. Figure 5a shows the molecular orbital diagram of CO_2_, along with its C_2_v symmetry. Among other catalysts, bismuth catalysts are known to be highly selective in the production of formate [71,72,73]. This characteristic of the bent configuration explains the affinity towards formate production. This is because the C_2v_ configuration, with the O end down, is more predisposed towards production of formate, as the overlap of both 1π_g_ and 2π_u_ orbitals with the metal states is favored by a linear geometry. Moreover, bismuth oxide, with high basicity phases, is able to engage in better bond formation due to the electronegative behavior of oxygen atoms [13].

The affinity for formate by bismuth-based catalysts can be also understood by the mechanism involved, specifically the intermediates and their adsorption behavior on the catalyst. The reaction pathways for CO_2_ reduction to formate can occur through three reaction intermediates: *COOH, *OCOH, and *H. Mostly, formate production is seen to prefer the *OCOH pathway, which is energetically the most favorable one [13]. Also, another report has reasoned the preference of formate generation by bismuth-based materials, as metals like Bi can convert the C of CO_2_ to HCOOH through protonation, due to the high overpotential for hydrogen evolution reaction and the weak adsorption of CO_2_^•−^ intermediate. This ardently favors the generation of formate over other carbon-based products with the use of bismuth-based catalysts [13,71].

Li et al., in 2018, synthesized nanostructured Bi electrocatalysts, where Bi nanoparticles formed a composite with Bi_2_O_3_ nanosheets. It was reported that the presence of abundant grain boundaries exposed highly active sites, which increased the selectivity of the products through stabilizing the reaction intermediates. A HR-TEM image of the catalyst showed that the grain boundaries were fabricated by some high-energy facets, like (012), (110), and (104). These high energy facets facilitated the CO_2_ reduction reaction, as well as the close contact of the Bi nanoparticles, which facilitated the charge transfer. This was proven with the Nyquist plots, where the composite of bismuth nanoparticles had a smaller diameter of a semicircle than bismuth nanosheets, thus proving lower charge-transfer resistance in the case of the nanoparticles. Bismuth nanoclusters confined into porous carbon were also reported to have FE of 96% for formate at −1.15 V vs. RHE. The confinement of the nanoclusters with the presence of the carbon matrix led to a decrease in the reaction energy barrier. Also, in situ Raman studies were carried out, and the key intermediate states during formate formation could be identified. From time-dependent in situ Raman studies at −1.15 V, it was seen that the two characteristic peaks of Bi (E_g_ and A1_g_) were unchanged, which confirmed the stagnant, stable presence of metallic Bi. In addition, the peak intensities in the case of the Raman studies of ν_as_CO_2_^−^ and *OCHO were seen to increase and decrease, conveying the fact of adsorption–desorption processes. The DFT calculations carried out demonstrated that the highly exposed (021) facet favored the generation of the *OCHO intermediate state [74]. The integration of graphene oxide into nanostructured Bi_2_O_3_ was also investigated by Melchionna et al. in 2022. The synthesis strategies in the work yielded reduced graphene oxide (rGO), incorporating Bi@Bi_2_O_3_ core–shell nanoparticles, and the other, where graphene oxide (GO) supported the oxidized Bi_2_O_3_ nanoparticles. The difference in the hierarchical structures was seen to affect the catalytic behavior. At −0.5 V vs. RHE, the HCOOH production rate by rGO/Bi@Bi_2_O_3_ equaled 3 ppm h^−1^ with an FE of 38%, while for GO/Bi_2_O_3_, the production rate equaled 73 ppm h^−1^ at −0.8 V vs. RHE with an FE of 46%. At −0.5 V vs. RHE, GO/Bi_2_O_3_ yielded low FE (11%) in comparison to its other counterpart. It was revealed that the Bi core was responsible for anticipating the potential of HCOOH formation, whereas the outer shell, consisting of the metal oxide, could be the cause of the modest current density, which in the case of GO/Bi_2_O_3_ was noted to be higher [75]. Another case of the very high selectivity (~100%) of formate production was observed, along with a high current density of 24.4 mA cm^−2^ by Bertin et al. in 2017 [76]. The work investigated in Bi and oxide-derived Bi films, which were prepared by potentiostatic electrodeposition on titanium substrates and subsequent electrochemical and thermal oxidation. The morphology of the electrochemically oxide-derived (EOD) Bi film was observed to be rougher in contrast to the flat structure observed prior to electroreduction. This rough morphology agreed with the electrochemical measurements, as the roughness factor for the as-deposited Bi was calculated to be 8 in comparison to the EOD Bi films, which was 30. It is to be understood that the term roughness factor (RF) is defined to be the ratio between the electrochemically active surface area (ECSA) and the geometric surface area. The main reduction products were H_2_ and formate. The Faradaic Efficiencies (FEs) for both the as-deposited Bi films and the EOD Bi films were similar; however, there was an increment in the current density in the case of the EOD-Bi films. This can be attributed to the increment in the electrochemical surface area, owing to its rough morphology [77]. In situ FT-IR studies can also be beneficial towards understanding the mechanism of eCO_2_RR. In one study, CuBi-MOF was used for eCO_2_RR, where the reaction mechanism of formate formation was studied over the catalyst surface. It was observed in the study that with an applied potential ranging from 0 to −1.2 V, the peak intensity of HCOO* at ~1379 cm^−1^ was seen to increase. It indicated the accumulation of HCOO* intermediate on the interface of the catalyst exceeding the consumption. This confirmed the HCOO* pathway to be preferable for formation production [78]. Zeng et al., in 2024, studied the activity of the bismuth-based catalyst according to the transformation undergone while in contact with the electrolyte KHCO_3_ under potential. At first, Bi_2_O_3_ polyhedral microcrystals were observed through SEM images, which transformed into a flower-like assembly of Bi_2_O_2_CO_3_ nanoflakes and then transformed into metallic bismuth. This catalyst was prepared by inkjet printing on the carbon paper, followed by thermal treatment. The FE for formate was noted to be 90% at −1.3 V vs. RHE [79]. Plasma-activated Bi_2_Se_3_ nanosheets were also tested to be selectively active for formate production, with FE > 90% at −1.2 V vs. RHE. These nanosheets with ultrathin structures provided a high surface area, which after a following electrochemical reduction transformed into porous Bi nanosheets [80].

Another interesting study was conducted by Gao et al. in 2019, where the group synthesized bismuth nanomaterials with different morphologies for the electrochemical reduction of CO_2_ to formate. The hydrothermal method was used to synthesize the three different morphologies. Bismuth nanowires, nanospheres, and nanosheets were formed, among which the nanosheets had the best electrochemical performance. This enhanced activity accounted for the exposed (012) facet with the highest texture co-efficient (TC), along with 10 nm of thickness of the nanosheets. At −0.85 V vs. RHE, Bi nanosheets showed formate production with an FE of 85%, whereas at the same potential the nanowires and nanospheres had an FE of 74% and 58%, respectively. Comparing the electrochemical activity between the nanospheres and nanowires, the nanowires had an increased activity. The TC value of (110) plane of nanowires was significantly larger than that of the nanospheres which, hence, was also explained to be a contributing facet in the electrochemical performance of the catalyst [81]. Shao et al., in 2018, investigated the different forms of Bi_2_O_3_ obtained by the calcination of Bi_2_O_2_CO_3_ at 300 °C, 400 °C, and 500 °C. The materials were drop cast on a glassy carbon electrode and then reduced to metallic Bi. The SEM images showed the change in morphology with temperature change, where an increase in temperature further than 400 °C led to the formation of nanocrystal aggregates from nanosheets. Among the different forms, the catalyst with the α form (calcination at 500 °C) performed the best, with 95% FE. This was definitive for the change in morphology and the phase of the Bi_2_O_3_, which ultimately had an effect on the electrochemical performance of the catalyst [82]. Thus, in one study, nanosheets had an improved activity over the other morphologies, and in another study it was seen that the aggregated nanosheets had a better performance comparatively. To understand this situation, we can look at another reported work that was carried out by Zheng and co-workers in 2021. A 2-D BiOI was synthesized and further reduced electrochemically to 1-D BiOI nanotubes. The catalyst was reported to be highly selective for formate, accounting its FE to be 97.1% with a partial current density of 31.1 mA cm^−2^ at an overpotential of only 790 mV. To understand the role of the specific morphology with their exposed active sites, DFT computations were carried out, and it was seen that the OCHO* intermediate preferentially adsorbed on the edge sites. When the edge sites were covered with oleylamine molecules, it resulted in a decrease in formate selectivity. Therefore, a morphological control allowing for exposed edges proved to increase the selectivity of the CO_2_ reduction reaction. Oleylamine was mainly used as a surfactant in the synthesis procedure, as in the absence of it, particles of a diameter of 1 µm were observed with a different crystal structure. In the presence of the surfactant, the architecture of 2-D nanosheets assembled from 1-D nanotubes.was formed. However, since the edge sites, without being covered with oleylamine, performed better in DFT studies, annealing was carried out to remove the surfactant. Interestingly, the morphology was still intact after the annealing treatment and, additionally, the fluffy morphology proved to expose the active sites far better [71]. Jiang et al., in 2022, electrochemically converted BiOCOOH nanowires into Bi/BiO_x_ nanosheets in situ at a constant potential of −0.91 V (vs. RHE) for 500 s or 1 h. SEM images portray the conversion of nanowires to nanosheets at 500 s, which further grow larger at 1 h. It was observed that the FE for formate of Bi/BiO_x_ nanosheets could reach ~94%, at a wide potential range from −0.78 V to −1.18 V vs. RHE, while Bi nanowires could attain only ~80% of FE [83]. Ning et al., in 2023, formed Bi_2_O_3_ microfibers with the assistance of cotton template through a heating treatment. As mentioned in the work, the helical structure of the Bi_2_O_3_ microfibers introduced lattice strains and oxygen vacancies, in addition to fully exposing the bismuth active sites. This enabled, in high FE, a ~100% formate production at −0.90 V vs. RHE. Also, in situ surface-enhanced Raman spectroscopy was carried out. It was seen that at −0.90 V vs. RHE, the characteristic peaks of Bi^3+^ at 313 and 447 cm^−1^ showed a gradual decrease with time, but along with that the characteristic peak of Bi^0^ at 141 cm^−1^ increased. Also, at 360 s, the Bi^3+^ peak was seen to completely disappear, which confirms the conversion of Bi_2_O_3_ to metallic Bi [84]. Thus, as can be seen, most reported works have accounted for higher catalytic performance in the case of 2-D nanosheet morphology, whether in clusters or present as sheets on the electrode surface. Table 4 summarizes the key points discussed in this section.

### 2.3. Photo-Electrocatalysis

In photo-electrocatalysis, one of the most important points to consider is the change in semiconductor band structure at the semiconductor-electrolyte interface. For example, in a p-type semiconductor, where the doped material is electron deficient when acting as a photocathode, it has its fermi level (E_F_) lower than that of the electrolyte solution. When the photocathode is immersed in the electrolyte, electrons flow from the electrolyte to the semiconductor, hence building an electric field in the semiconductor near the interface. This creation of the electric field results in the band bending of the semiconductor’s electronic structure. In the case of a p-type semiconductor, the semiconductor’s valence band (VB) and conduction band (CB) will bend downwards, thus applying a force on the holes and electrons in the opposite direction. This will drive them towards/away from the interface, hence enhancing the charge separation [10] (Figure 6).

Facet engineering and morphology control can have an effect on the overall photoelectrochemical performance of the catalyst. A good example of facet engineering for enhanced photoelectrochemical performance can be derived from the work of Wang and co-workers. A change in the exposed crystal plane (101) of BiOCl was noted when forming a heterojunction with Bi_2_WO_6_. The exposed crystal plane of the pristine form changed to (112) in the heterojunction, which had good compatibility with the (113) plane of Bi_2_WO_6_. This imparted better photocurrent densities of the composite as compared to the pristine BiOCl, as was evident from the transient photocurrent curves [86]. The proper separation of the charge will inhibit the recombination of electrons and holes, which would vary with different values of bandgaps. These bandgaps depend a lot on the specific morphology of the bismuth material. A list of various morphologies of bismuth-based materials and their band gaps involved in catalytic CO_2_ reduction are tabulated below in Table 5.

Hence, as can be observed from Table 5, altering the morphology of the same material can help in tuning the bandgap, which ultimately will have a drastic effect on the photoelectrochemical performance of the catalyst with a band bending effect and the appropriate separation of charge carriers.

Metallic bismuth-modified materials have also been used in the photo-electrocatalysis of CO_2_. In one of the reports, it was seen that the separation and migration of the photogenerated holes and electrons were enhanced with Bi doping in one-dimensional ZnO/α-Fe_2_O_3_ nanotubes due to the presence of n-n heterojunction. It also led to the increase in its conductivity as compared to the single semiconductors. An HR-TEM analysis showed (003) as the active facet of metallic Bi [93]. Metallic bismuth doping was also investigated on ZnO/planar-Si, where bismuth was electrodeposited and then, upon drying overnight at 80 °C, resulted in a Bi/Bi_2_O_3_ form. The electrodeposition time played an important role in the performance of the catalyst, as it led to the deposition of bismuth with different morphologies. Interestingly, the major product formed was formate, which had a volcano trend over the increment in deposition time of bismuth. The morphology of the bismuth species changed from dendritic in the first one minute of the deposition time to thin nanosheets with lose and porous morphology in 5 min time. Further extending the deposition time to 10 min, thick layers of bismuth species were formed, which covered the Si substrates. Due to the change in morphology (Figure 7), the formate selectivity was found to lessen at 10 min as compared to the selectivity offered by the nanosheets formed at 5 min. Moreover, the comparison of photoelectrochemical CO_2_RR (PEC) and the electrochemical CO_2_RR (EC) reduction activity of CO_2_ was carried out at −0.95 V vs. RHE, and it was seen that formate was the main reduction product obtained with PEC, whereas both formate and hydrogen were the main products in the case of EC [94].

Bismuth materials, when formed into a heterojunction with other bismuth-based materials, can alter their active facet and, hence, change their catalytic activity. On forming a heterojunction of BiOCl with Bi_2_WO_6_, the exposed crystal plane of BiOCl changed from (101) of the pristine form to (112) in the heterojunction. It was reported that at −0.6 V, as shown in Figure 8a, the pristine material (BiOCl) formed minimal amounts of methanol, whereas the heterojunction-based catalyst prepared with hydrothermal synthesis for 10 h (BCW-10) produced methanol at the rate of 12.5 µM h^−1^ cm^−2^. However, the catalyst synthesized with 6 h of hydrothermal treatment (BCW-6) produced ethanol at a rate of 11.4 µM h^−1^ cm^−2^. This change in selectivity of the product evidently depended on the change in morphology of the catalyst with the increment in the treatment time. The thickness of the heterojunction increased to 15 µm in the case of BCW-6, while maintaining the layered structure (Figure 8c,d) [86].

Ding and co-workers prepared Si/Bi interfaced photocathodes, which had a high performance of formate production. The amount and morphology of Bi deposits on the surface of Si were controlled by the reaction time. A reaction time of 1 min resulted in the formation of Bi nanoparticles, whereas a reaction of ~7–30 min formed nanoflakes. This change in morphology affected their performance in the selectivity of products, as Si/Bi (1 min) suffered from severe parasitic H_2_ evolution reaction, with less than 40% of formate selectivity, while the photocathode with nanoflake morphology improved its performance by 60–90% [95]. There have been several works where BiVO_4_ was used as a photoanode [96,97,98,99,100,101]. Kang et al., in 2021, studied a PEC cell with reduced graphene oxide layered TiO_2_ as the photocathode and (040) crystal facet engineered BiVO_4_ as the photoanode. The same group, in 2016, investigated the specificity of the (040) crystal facet in BiVO_4_. The photocurrent density of the photoanode under AM 1.5G illumination was reported to be 0.94 mA cm^−2^ and produced 42.1% of the absorbed photon-to-current conversion efficiency at 1.23 V vs. RHE. Mainly, the solar light conversion efficiency is directly proportional to the product of the solar light absorption efficiency, charge separation efficiency and surface charge transfer efficiency. Therefore, the enhanced PEC performance of the BiVO_4_ nanoplates was due to the interfacial electron transport reaction between the {010} plane and the electrolyte, with better charge separation efficiency and surface charge transfer efficiency [102,103]. Ren et al. prepared a heterojunction-based Bi_2_S_3_ nanoflowers and ZIF-8 composite, where ZIF-8, the porous crystal material, was responsible for capturing and activating CO_2_ molecules. Bi_2_S_3_ carried out the photoelectrocatalytic CO_2_ reduction to formate at the potential of −0.7 V vs. RHE under visible light. The FE of formate was 74.2%, and the maximum current density was reported to be 16.1 mA cm^−2^. It was highlighted in this work how the morphological attributes of the flower structure of Bi_2_S_3_ avoided the ZIF-8 surface stacking structure and optimized the electronic transition path. This enabled the catalytic material to receive more photonic energy and improve the photocurrent density and efficiency of the photoelectrocatalytic reaction [104]. BiVO_4_ in the cathodic compartment was investigated in a PEC cell for CO_2_ reduction reactions. The obtained products were methanol and acetic acid, with production values of 22 and 5.5 µmol cm^−2^, respectively. The improved activity of the catalyst, despite having poor photocatalytic activity, was attributed to the formation of metallic Bi sites on the surface of BiVO_4_. It enabled charge transfer, reducing the recombination of the charge carriers in addition to the application of an external potential bias in the case of photoelectrochemical CO_2_RR [105].

## 3. Role of Facets and Morphology in Adsorption of CO_2_

The surface properties and morphology of the catalysts affect the adsorption of CO_2_, which play an eminent role in the catalytic reduction reaction of CO_2_. Oxygen vacancies, which are pervasive point defects in metal oxides, can play a crucial role in the adsorption of the CO_2_ molecule. Miao et al. synthesized Bi_2_O_3_ in two different morphologies of nanoparticles and nano-thin rod clusters. The nanoparticles accounted for more catalytic activity in comparison to the nano-thin rods, as the former had more cavities and edges. It can be understood that exposure to more cavities and edges allowed for the better interaction of the CO_2_ molecules with the oxygen vacancies, thus improving the kinetics of the reaction. XPS analysis showed more intense peaks for the nanoparticles as compared to the nanorods, at 285.3 and 535.8 eV, indexed at C1s and O1s, respectively. These intense peaks can be due to the presence of oxygen vacancies, allowing for the better adsorption of the CO_2_ molecules. Hence, the current density of CO_2_ reduction at −1.2 V vs. RHE for the nanoparticles could reach ~22.4 mA cm^−2^, while the nanorods could only reach ~8.8 mA cm^−2^ [106]. There have been many reports where the photocatalytic reduction of CO_2_ had boosted performance due to the exposed (001) facet in the catalyst [68,107,108,109]. This is because the (001) plane of the metal oxide can form oxygen vacancies [110]. Kong et al. synthesized heterojunction-based 2-D nanosheets of BiOI and oxygen-deficient Bi_2_WO_6_, with {001} exposed facets on both the top and bottom surfaces. The 2-D nanosheet morphology of both the semiconductors allowed for strategic coupling with the highly reactive {001} facet. This favorable 2-D morphology and the enhanced coupling of the two semiconductors improved the charge transfer across the heterojunction. The oxygen vacancies present on the exposed facet served as active sites for CO_2_ adsorption and activation. The CO_2_ molecule, hence, was activated by accepting electrons from the oxygen vacancies and formed CO_2_^•−^ radicals, a step that represents the rate-limiting step of CO_2_ reduction [111].

Despite numerous works and reports on oxygen vacancies proving to be beneficial for CO_2_ reduction reactions, there are some reported works where it proved otherwise. CO_2_ has a soft acidic nature and the presence of strong basicity phases like oxides that can increase the adsorption of CO_2_ molecules [112]. Ma and co-workers, in 2022, found out that Bi-O moieties in Bi-based catalysts can improve the rate-limiting electron process. Bi_2_O_2_O/Bi_2_O_2_(OH)(NO_3_), referred to as BiON in the work, was synthesized in three different sheet-like morphological patterns. Ultrathin nanosheets, stacked sheet particles and flat nanosheets were subjected to electrochemical CO_2_ reduction with a 1 h chronoamperometry test at −1.0 V vs. RHE. XRD patterns were observed on the spent catalysts, and the ultrathin nanosheets were still seen to be intact with Bi-O moieties, whereas the other two were completely reduced into the metallic form. These ultra-thin nanosheets performed better as catalysts, with great stability and with faradaic efficiency for formate production, equaling to 98%. It was found that the adsorption energies of radical intermediates, CO_2_^•−^ and OCHO^•^, on Bi_2_O_2_O were calculated to be 5.17 and 1.19 eV, respectively. However, the adsorption energies of CO_2_^•−^ and OCHO^•^ on metallic Bi (001) were calculated to be 2.64 and 0.86 eV, respectively. The higher adsorption energies indicated a more stable adsorption site for the intermediates on Bi-O moieties as compared to the metallic form [83]. Deng et al. synthesized metallic bismuth samples with spherical morphology and a smooth surface. The samples showed a characteristic (012) plane. An oxidation treatment was carried out, which introduced some stress fractures and shape deformations and the formation of Bi-O structures. With this, β-phase Bi_2_O_3_ was observed by HR-TEM with the exposed facet of (201). This resulted in improved effects in the selectivity and activity of CO_2_ reduction to formate. Hence, the Bi sphere catalyst presented a partial current density of only ~1.5 mA cm^−2^, while Bi_2_O_3_ showed ~8 mA cm^−2^ at −0.9 V vs. RHE for formate production [113]. Walker et al., in 2015, studied pyrochlore compounds, which have the nominal composition A_2_B_2_O_7_. These compounds have two distinguishable oxygen sites, often expressed by the stoichiometry A_2_B_2_O_6_O’. Cubic bismuth complexes were prepared, which possess high CO_2_ chemisorption capacity. The high surface basicity can be associated with O 2p-Bi 6s6p hybridized electronic states. These electronic states are more favorable to donating electronic density to adsorbed species than the surface lattice oxygen ions. Also, it was noted that the overall morphology tuning into cubic oxides let the BiO_x_ termination increase the density of basic surface sites. This enabled stronger adsorption of CO_2_ [114].

Hierarchical and porous structures are better adsorbents of CO_2_ molecules. As seen in Figure 9, higher magnification microscopy enables us to see the detailed surface structures of the Bi_2_WO_6_ microspheres. Crossed nanosheets and broken microstructures with hollow openings can be seen. This provides better capability for Bi_2_WO_6_ hollow microspheres to adsorb CO_2_ molecules than that of bulk Bi_2_WO_6_, as proven in the adsorption isotherm [70]. The adsorption capacity of CO_2_ can also determine the fate of the reaction and determine the reaction pathway. A study was carried out by Wang et al. in 2022 on the lattice dislocation of Bi catalysts, where special adsorption behavior and chemical activity for reactants and reaction intermediates was observed, which specifically influenced the CO_2_RR path over the electrocatalyst surface. Lattice-perfect Bi (P-Bi) and lattice-dislocated Bi (LD-Bi) were synthesized with the electrochemical reduction of Bi_2_O_2_CO_3_. Both had a lamellar nanosheet structure, with LD-Bi exhibiting a thinner nanosheet structure, which could be attributed to the electrochemical stripping at high reduction current. Chronopotentiometric measurements led to obtaining clear evidence that LD-Bi, with lattice dislocations, enhanced the CO_2_RR process more as compared to the P-Bi throughout the potential range of −0.57 to −1.27 V vs. RHE. Moreover, the partial current density of P-Bi was measured to be less than 20 mA cm^−2^ throughout the entire potential range, while that of LD-Bi was measured to be more than 50 mA cm^−2^ at −1.27 V vs. RHE. It was discussed in the work how the lattice dislocations played an important role in the improved performance of the catalyst, relating to the adsorption behavior of CO_2_ (Figure 10a). LD-Bi could provide more adsorption sites for CO_2_ reduction, which facilitated the kinetics of electrochemical reduction reactions to formate. Considering OH^−^ as a substitute for CO_2_*, an oxidation Linear Sweep Voltammetry (LSV) curve study was carried out to evaluate the adsorption capacity of CO_2_*. The behavior of OH^−^ on the surface of LD-Bi and P-Bi was studied under N_2_ saturated 0.1M KOH solution. LD-Bi exhibited a stronger negative peak, as can be seen in Figure 10b, indicating a stronger adsorption capacity for CO_2_* on LD-Bi. CO_2_* forms a strong bond with H^+^ to form OCHO* intermediate, which thus forms HCOOH with high selectivity [71].

## 4. Conclusions

The catalytic route for CO_2_ reduction with bismuth catalysts is favorable in its various morphologies and forms, as has been reported widely in the scientific literature, in addition to modification of the catalyst by introducing defects or forming a heterojunction with other semiconductors. With this review, it can be understood how engineering the catalysts to specific facet exposure can enhance the performance of the catalyst. This review analyses the various cases reported in the literature involving bismuth-based materials and compares the activity of the catalyst in its CO_2_ reduction behavior. This analysis is based on the activity of the catalysts relating to their exposed facets and differences in morphology, which could allow us to better understand how to synthesize the materials with preferential facet exposure to improve the charge transfer properties of the materials, and adsorption of reactants/intermediates on the material surface.

Among the oxyhalides, it was seen how {001} facet dominant catalysts had a higher photocatalytic activity due to the internal electric field being directed in the [001] direction. Moreover, enhanced charge transfer properties were noted for (010) facet on depositing Bi nanoparticles, which further encourages the fact that properties can be modulated by the modification of specific facets. In the case of BiOBr, it was realized that introducing defects could facilitate the exposure of the active facet, which would improve the photocatalytic activity. Also, the heterojunction formation of BiOBr, having (110) as the exposed facet with Indium Oxide, improved its photocatalytic properties. In the case of Bi_2_O_3_, where the metal-oxide can exist in various forms, the monoclinic phase and β phase were studied due to the limited work available in the literature for bismuth oxide-based materials. For monoclinic Bi_2_O_3_ formed into a heterojunction with CuBi_2_O_4_, the (020) facet exposure added in the enhanced H_2_O oxidation, thus facilitating the kinetics of the reaction. In the case of the β phase, the (201) facet exposure of Bi_2_O_3_ was reported in the work to enhance the composite formation with g-C_3_N_4_, which enhanced the better separation of charges, thus improving photocatalytic properties. The modulation of the behavior of the catalyst was also noted on the formation of the Schottky junction, where the composite BiVO_4_{010}-Au-Cu_2_O performed better than BiVO_4_{110}-Au-Cu_2_O. It was also seen how maintaining the appropriate proportion of the exposed {010} and {100} facets increased the CO evolution rate in the case of BiOIO_3_. Also, nanostrips of BiOIO_3_, which had optimal oxygen vacancy concentration, performed better than BiOIO_3_ nanoparticles, which shows the benefit of morphology control. For 1-D Bi_2_S_3_, to improve the photocatalytic properties, strategies like uniform dispersion are beneficial, along with also the formation of a heterojunction. Moreover, morphology played an important role, since, the comparison of the activity of Bi_2_S_3_ in the case of urchin-like morphology was noted to be higher as compared to the nanoparticles. Theoretical studies conveyed the fact that CO_2_ dissociation in the case of Bi_2_WO_6_ exposed with {001} facet is easier as compared to the {010} and {101} facets. Also, Bi_2_WO_6_, morphology played an important role, with its nanoflakes being more feasible to form into a heterojunction with g-C_3_N_4_.

The electrocatalytic reduction of CO_2_ mainly forms formate, with gaseous products consisting of H_2_ and CO in the case of bismuth-based materials. In the reported works discussed so far, exposure to high energy facets, adsorption properties based on specific morphologies, a roughness factor adhering to increases in the performance of the catalyst has been observed. Morphological control, which led to exposed edges, increased the selectivity of the CO_2_RR. The composite formation of bismuth nanoparticles with Bi_2_O_3_ nanosheets had better performance than when forming the composite of bismuth nanosheets with Bi_2_O_3_, which was attributed to abundant grain boundaries that were fabricated by some high energy facets. However, in another case, it was seen that when only bismuth nanostructures were considered, the highest performing material was bismuth nanosheets, followed by bismuth nanowires and then nanoparticles. Therefore, the effect of composite formation can be seen here, where alterations in the charge transfer of bismuth nanostructures can occur by the formation of composites. Moreover, another factor to take into account is the electrochemical surface area, where evidently in one report it was seen that the roughness factor aided in higher current densities. Also, Bi_2_O_3_ microfibers, formed through cotton template in one report, introduced lattice strains and oxygen vacancies along with exposing the bismuth active sites. This resulted in FE of ~100% formate production at −0.90 V vs. RHE. Apart from experimental procedures, DFT studies could help in understanding the preference of the active sites. In one study it was shown how OCHO^*^ intermediate preferred to adsorb on the edge sites. More DFT computational studies including the different morphologies of bismuth-based materials exposing different facets could help us to know further on this aspect.

Photoelectrocatalysis by various bismuth-based catalysts was considered, where the importance of band-gap modulation was seen, as it was dependent on the different morphologies of the respective catalysts. The variation in product selectivity was also seen in the case of bismuth deposition, where the morphology of the catalysts changed with changes in deposition time and consecutively affected the selectivity of formate. In this case, the nanosheets offered a higher amount of formate as compared to the thick layers of bismuth species. The importance of photoelectrochemical reduction instead of a solely electrochemical reduction was realized, as the former led to the formation of only formate while the latter led to a mixture of formate and hydrogen as the main products. More of the reported works discussed in this review enable us to see how the morphological differences in the catalysts affects the product selectivity a whole lot more than photocatalysis or electrocatalysis. An example was seen in one of the reported works, where the pristine BiOCl formed minimal amounts of methanol; however, in forming a composite with Bi_2_WO_6_, it produced higher amounts of methanol in the case of a synthesis time of 10 h. But, the effect of morphology change and product selectivity was realized when the composite produced higher amounts of ethanol when changing the synthesis time to 6 h. Also, the use of BiVO_4_ has been seen to be used extensively as photoanodes, where the exposure of active facets also facilitates the charge-transfer processes, thus benefitting them for the PEC cells.

The adsorption of CO_2_ and its dependence on facets and morphologies was understood with the exposure of cavities and edges, the inclusion of oxygen vacancies, and the modulated interaction of CO_2_ molecules with the oxygen vacancies. Exposure to more cavities and edges allowed for the better adsorption of the CO_2_ molecules, allowing for a better interaction with the oxygen vacancies that could improve the kinetics of the reaction. In the previous section of photocatalytic CO_2_ reduction reactions, we saw how the exposure of the (001) facet in maximum cases of bismuth oxyhalides played an important role in the improvement of their activity. This is because adsorption of CO_2_ is realized to be facilitated with (001) exposed facet, which would contain oxygen vacancies. However, alternating conclusions were observed, where some articles stated, upon introduction of Bi-O moieties, to have increased the CO_2_ adsorption. It was reported that overall morphology tuning into cubic oxides could let the BiO_x_ termination increase CO_2_ adsorption. Further research on this aspect is required to derive a definite conclusion. Also, it was understood that hierarchical and porous structures are better adsorbents for CO_2_ molecules.

Hence, multiple factors come to light to have an overall grasp on the role played by the modulation of the morphology, as well as the exposure of the active facets. To synthesize a particular bismuth-based material, and aim to improve the charge transfer process or to improve the selectivity towards the production of a specific product, it could be beneficial to consider such synthesis strategies of modulating facets and morphologies. However, it can be realized that to have a deeper understanding, more theoretical studies would be a definite positive addition in broadening the knowledge in this aspect. It can be understood that, overall, a careful understanding of the proper choice of the catalysts, with appropriate exposed planes for obtaining a selective and higher quantity of products, would be strategically advantageous.

## Figures and Tables

**Figure 1 materials-17-03077-f001:**
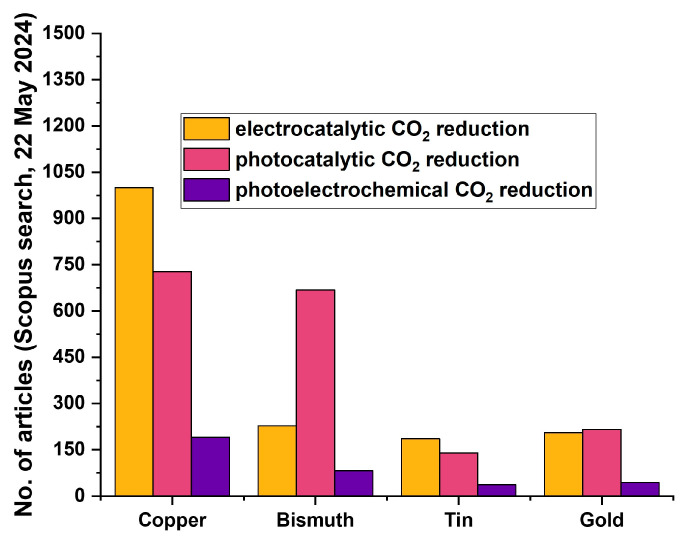
Graphical representation of the statistical report of search results in Scopus database, based on keywords relevant to this review.

**Figure 2 materials-17-03077-f002:**
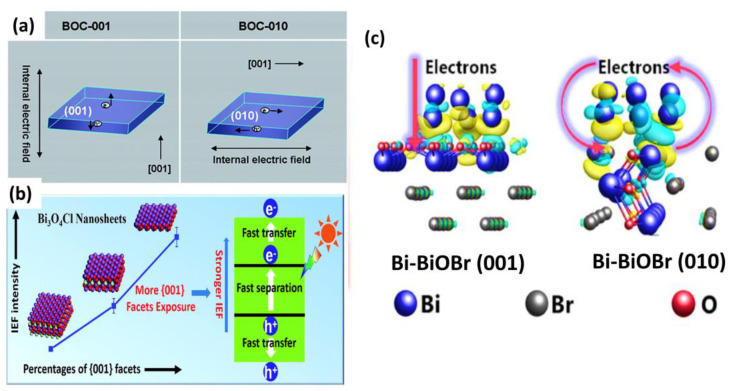
Representation of different charge transfer routes (**a**): Direction of Internal Electric Field (IEF) in Bi_x_O_y_Cl with (001) and (010) facets exposed and (**b**) Photocatalytic activity dependent on IEF of Bi_3_O_4_Cl nanosheets. Reproduced from reference [46] with permission from The Royal Society of Chemistry, 2014. (**c**) Schematic illustration of a new charge transfer route established between the contacted Bi NPs and the (010) facet of BiOBr. Reproduced from reference [48] with permission from Elsevier 2018.

**Figure 3 materials-17-03077-f003:**
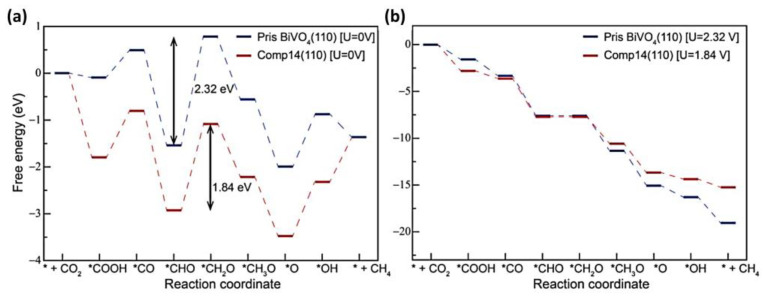
Free energy diagrams for CO_2_RR on pristine BiVO_4_ (110) and comp14 (110) at (**a**) 0 V and pH = 0 and at (**b**) U = specified potential (barrier for corresponding PDS) values and pH = 13. The asterisks denote the adsorption sites for the reaction intermediate in the eCO_2_RR process. Reproduced from reference [53] with permission from the American Chemical Society, 2022.

**Figure 4 materials-17-03077-f004:**
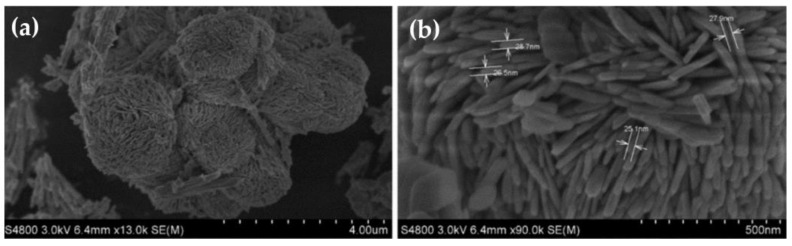
SEM images of (**a**) Bi_2_S_3_ microspheres (**b**) Bi_2_S_3_ microspheres with packed nanoplates. Reproduced from reference [66] with permission from The Royal Society of Chemistry, 2019.

**Figure 5 materials-17-03077-f005:**
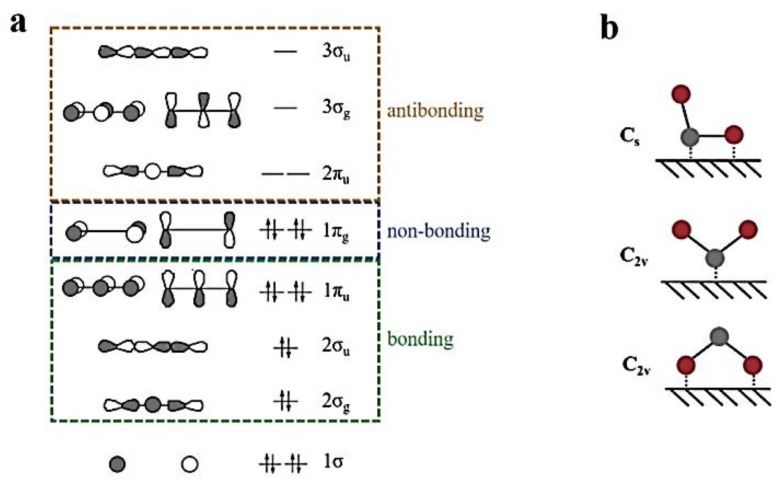
Molecular orbital diagram and possible coordination of CO_2_ on catalyst surface (**a**): Energy levels for molecular orbitals of CO_2_. (**b**) Possible coordination modes of CO_2_ on a metal surface. The red and grey circles denote oxygen and carbon atoms respectively. Reproduced from reference [13] with permission from Elsevier 2020.

**Figure 6 materials-17-03077-f006:**
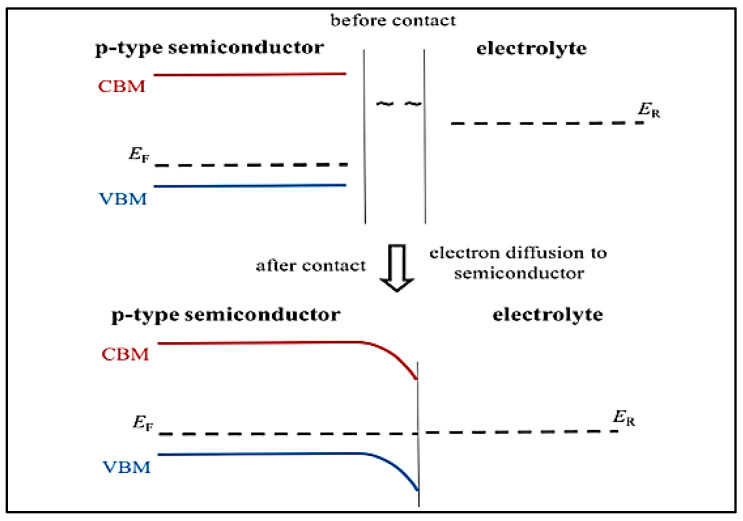
Schematic of band bending at the semiconductor/electrolyte interface. Reproduced from reference [10] with permission from American Chemical Society, 2019.

**Figure 7 materials-17-03077-f007:**
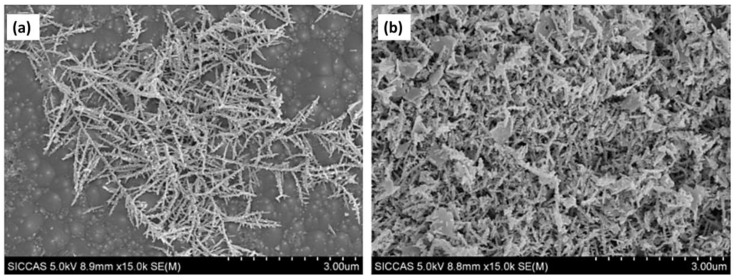
SEM images of Bi-Bi_2_O_3_/ZnO/p-Si for different deposition times (**a**) 1 min and (**b**) 10 min. Reproduced from reference [94] with permission from The American Chemical Society, 2022.

**Figure 8 materials-17-03077-f008:**
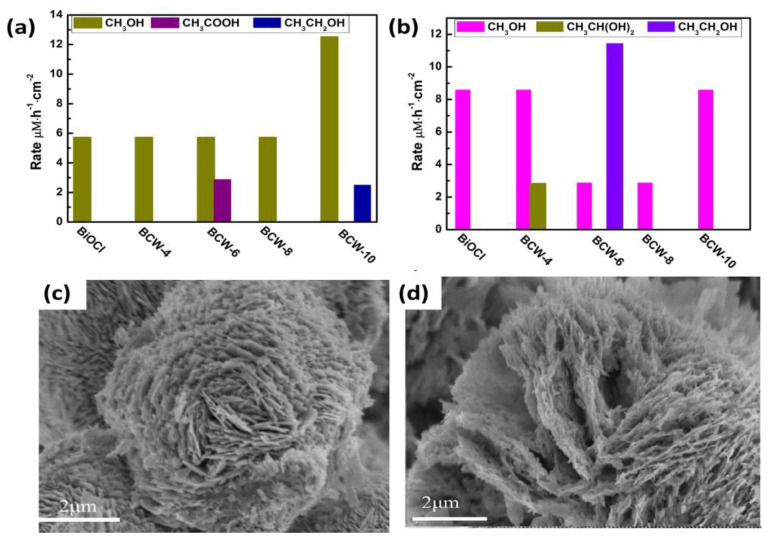
The production rates of heterojunction PEC cells for five different photocathodes at (**a**) −0.6 V and (**b**) −1.0 V; SEM images of (**c**) BCW-10 and (**d**) BCW-6. Reproduced from reference [86] with permission from Elsevier, 2019.

**Figure 9 materials-17-03077-f009:**
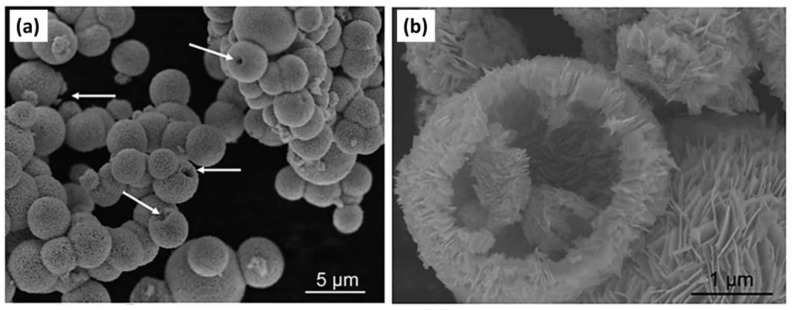
SEM images (**a**) Bi_2_WO_6_ microspheres with the arrows showing hollow opening and (**b**) magnified hollow microstructures of Bi_2_WO_6_. Reproduced from reference [70] with permission from The Royal Society of Chemistry, 2019.

**Figure 10 materials-17-03077-f010:**
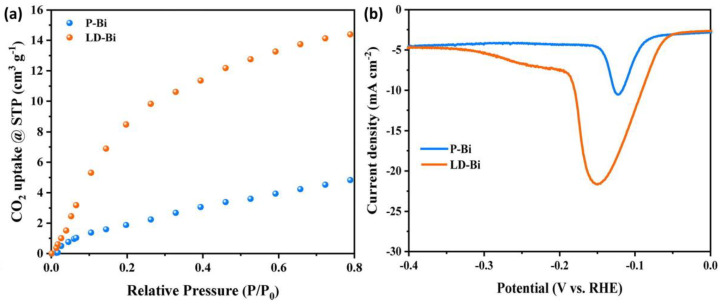
(**a**) CO_2_ adsorption isotherm of P-Bi and LD-Bi. (**b**) Oxidative LSV curves of P-Bi and LD-Bi in N_2_-saturated 0.1 M KOH. Reproduced from reference [71] with permission from Elsevier, 2022.

**Table 1 materials-17-03077-t001:** Standard Electrochemical Potential (E^0^ vs. RHE) for CO_2_ reduction reactions under standard conditions (1 atm, 25 °C) ^1^.

Reaction	E^0^ vs. SHE(V)
Half-electrochemical thermodynamic reactions	Electrode potentials (V vs. SHE) under standard conditions
CO_2_ (g) + e^−^ → CO_2_^•−^ (aq)	−1.990
CO_2_ (g) + 2 H^+^ + 2e^−^ → HCOOH (l)	−0.250
2 CO_2_ (g) + 2 H^+^ + 2e^−^ → H_2_C_2_O_4_ (aq)	−0.500
CO_2_ (g) + 2 H_2_O (l) + 2e^−^ → HCOO^−^ (aq) + OH^−^	−1.078
2 CO_2_ (g) + 2e^−^ → C_2_O_4_^2−^ (aq)	−0.590
CO_2_ (g) + 2 H^+^ + 2e^−^ → CO (g) + H_2_O (l)	−0.106
CO_2_ (g) + 2 H_2_O (l) + 2e^−^ → CO (g) + 2 OH^−^	−0.934
CO_2_ (g) + 4 H^+^ + 4e^−^ → C (s) + 2 H_2_O (l)	0.210
CO_2_ (g) + 2 H_2_O (l) + 4e^−^ → C (s) + 4 OH^−^	−0.627
CO_2_ (g) + 3 H_2_O (l) + 4e^−^ → CH_2_O (l) + 4 OH^−^	−0.898
CO_2_ (g) + 6 H^+^ + 6e^−^ → CH_3_OH (l) + H_2_O (l)	0.016
CO_2_ (g) + 5 H_2_O (l) + 6e^−^ → CH_3_OH (l) + 6 OH^−^	−0.812
CO_2_ (g) + 8 H^+^ + 8e^−^ → CH_4_ (g) + 2 H_2_O (l)	0.169
CO_2_ (g) + 6 H_2_O (l) + 8e^−^ → CH_4_ (g) + 8 OH^−^	−0.659
2 CO_2_ (g) + 12 H^+^ + 12e^−^ → CH_2_CH_2_ (g) + 4 H_2_O (l)	0.064
2 CO_2_ (g) + 12 H^+^ + 12e^−^ → CH_3_CH_2_OH (l) + 3 H_2_O (l)	0.084
2 CO_2_ (g) + 9 H_2_O (l) + 12e^−^ → CH_3_CH_2_OH (l) + 12 OH^−^	−0.744

^1^ Data obtained from reference [11].

**Table 3 materials-17-03077-t003:** Comprehensive table with a summary of the photocatalytic CO_2_ reduction reactions.

Material, Exposed Facet and/or Morphology	Products Evolved	Conditions	Reference
BiOBr, {001}	CO, 4.45 µmol g ^−1^ h^−1^	Closed gas reactor, CO_2_, Simulated sunlight, 20 °C	Wu et al., 2017 [42]
IO/BiOBr, heterojunction of (110) BiOBr and (222) IO	CO, 54.2 µmol g^−1^	50 mg + 10 mL H_2_O + CO_2_, 60 °C, 300 W Xe lamp	Liu et al., 2024 [50]
CuBi_2_O_4_/Bi_2_O_3_, where Bi_2_O_3_ with (020)	CO (247.62 µmol m^−2^), CH_4_ (119.27 µmol m^−2^), O_2_ (µmol m^−2^)	Gas–solid catalytic system, CO_2_ + H_2_O vapor, λ > 400 nm	Shi et al., 2021 [51]
40% Bi_2_O_3_/g-C_3_N_4_, Bi_2_O_3_ with (201)	CO (22.5 µmol g^−1^)	50 mg + 100 mL H_2_O + CO_2_, 300 W Xe lamp, 25 °C	Peng et al., 2019 [52]
BiVO_4_/WO_3_, BiVO_4_ with (110)	CH_4_ (105 µmol g^−1^ h^−1^)	5 mg + 30 mL 0.1 M NaOH, λ > 400 nm	Das et al., 2022 [53]
BiVO_4_ (monoclinic, scheelite)	CH_3_CH_2_OH (406.6 µmol h^−1^)	200 mg + 100 mL H_2_O + CO_2_, 300 W Xe lamp, 0 °C	Liu et al., 2009 [54]
BiVO_4_ (tetragonal, zircon)	CH_3_CH_2_OH (4.9 µmol h^−1^)	200 mg + 100 mL H_2_O + CO_2_, 300 W Xe lamp, 0 °C	Liu et al., 2009 [54]
BiVO_4_{010}-Au-Cu_2_O	CO (2.02 µmol g^−1^ h^−1^), CH_4_ (3.14 µmol g^−1^ h^−1^)	100 mg + 0.4 mL H_2_O + CO_2_, 300 W Xe lamp, λ ≥ 400 nm	Zhou et al., 2018 [56]
BiVO_4_{110}-Au-Cu_2_O	CO (1.12 µmol g^−1^ h^−1^), CH_4_ (1.22 µmol g^−1^ h^−1^)	100 mg + 0.4 mL H_2_O + CO_2_, 300 W Xe lamp, λ ≥ 400 nm	Zhou et al., 2018 [56]
BiOIO_3_ {010}/{100}	CO (5.42 µmol g^−1^ h^−1^)	50 mg, 1.70 g NaHCO_3_,15 mL H_2_SO_4_, 300 W Xe lamp	Chen et al., 2018 [59]
BiOIO_3_ (bulk)	CO(1.77 µmol g^−1^ h^−1^)	50 mg, 1.70 g NaHCO_3_,15 mL H_2_SO_4_, 300 W Xe lamp	Chen et al., 2018 [59]
Bi_2_O_2_CO_3_/BiOIO_3_	CO (224.71 µmol g^−1^ h^−1^)	35 mg + 70 mL H_2_O + CO_2_, 300 W Xe lamp (1.5 AMG), 25 °C	Zhang et al., 2024 [61]
20% Bi_2_S_3_/g-C_3_N_4_ (Bi_2_S_3_ as quantum dots distributed over g-C_3_N_4_)	CO (6.84 µmol g^−1^ h^−1^), CH_4_ (1.57 µmol g^−1^ h^−1^), H_2_ (1.38 µmol g^−1^ h^−1^)	50 mg + 100 mL H_2_O + CO_2_, 300 W Xe lamp	Guo et al., 2020 [64]
Bi_2_S_3_, nanoparticles; Bi_2_S_3_ urchin-like spheres; Bi_2_S_3_ microspheres; thin urchin-like Bi_2_S_3_ spheres	Methyl formamide (300 µmol g^−1^; ~460 µmol g^−1^; 700 µmol g^−1^; 350 µmol g^−1^)	10 mg + 10 mL CH_3_OH + CO_2_, 25 °C, 250 W high-pressure Hg lamp. This is followed by esterification reaction to obtain methyl formamide	Chen et al., 2013 [66]
Bi_2_WO_6_ nanoplates with {001} exposed facet	CH_4_ (1.1 µmol g^−1^ h^−1^)	100 mg in a gas-enclosed quartz reactor, CO_2_, λ > 420 nm	Zhou et al., 2011 [68]
Bi_2_WO_6_/g-C_3_N_4_ with Bi_2_WO_6_ nanoflakes	CO (5.19 µmol g^−1^ h^−1^)	100 mg in a gas-closed circulation system with CO_2_ + H_2_O, 300 W Xe lamp, λ > 420 nm, 25 °C	Li et al., 2015 [69]
Bi_2_WO_6_ hollow microspheres; Bi_2_WO_6_ bulk	CH_3_OH (32.6 µmol g^−1^; 1.28 µmol g^−1^)	200 mg + 100 mL H_2_O + CO_2_, 300 W Xe lamp, λ ≥ 400 nm, 0 °C	Cheng et al., 2012 [70]

**Table 4 materials-17-03077-t004:** Comprehensive table with a summary of the electrocatalytic CO_2_ reduction reactions.

Material	Synthesis	Potential (V vs. RHE) and FE_HCOO_^−^	Ink Deposition Method	Reference
Bismuth nanocluster confined into porous carbon	BiCl_3_ solution dropped into porous carbon solution, followed by 24 h stirring and drying at 80 °C and annealing at 600 °C for 1 h	−1.15 V, 96%	100 µL of ink evenly spread on 1 × 1 cm^2^ carbon paper, dried with infrared lamp	Yu H. et al., 2023 [74]
Bi nanoparticles/Bi_2_O_3_ nanosheets	Hydrothermal method	−0.76, 100%	Ink paste deposited into glassy carbon electrode of 3 mm diameter	Li L. et al., 2019 [76]
Bi and oxide-derived Bi films	Electrodeposition on Ti substrates at different positive potential limits (2.7–10.7 V)	−0.85, ~80%	Electrodeposition forming 1.1 µm thick Bi film.	Bertin E. et al., 2017 [77]
Flower-like assembly of Bi_2_O_2_CO_3_ nanoflakes reduced to metallic Bi	Ink used for inkjet printing consisted of bismuth nitrate pentahydrate, water, and nitric acid.	−1.3 V, 90%	Inkjet printer used on carbon paper	Zeng et al., 2024 [79]
Bi_2_Se_3_ nanosheets to porous Bi nanosheets	Plasma assisted method	−1.2 V, >90%	Ink solution dripped in carbon cloth and dried naturally	Xiao et al., 2024 [80]
Bi nanosheets	Hydrothermal method	−0.85, 85%	100 µL of ink evenly spread on 1 cm^2^ carbon paper, dried with infrared lamp.	Gao T. et al., 2019 [81]
Bi nanowires	Solvothermal with ethylene glycol as solvent	−0.85, 75%	-do-	-do-
Bi nanospheres	Solvothermal with ethylene glycol and acetone in the ratio of 1:2	−0.85, 55%	-do-	-do-
Metallic Bi	Metallic Bi formed by reduction from Bi_2_O_3_ formed from calcining Bi_2_O_2_CO_3_ at 500 °C	−0.96, 95%	10 µL of ink cast on 5 mm diameter L-style glassy carbon electrode	Shao L. et al., 2019 [82]
Metallic Bi	Metallic Bi obtained by electrochemical reduction of BiOI	−0.90, 97%	Ink drop cast on carbon paper (0.5 × 1.0 cm^2^) with 2 mg/cm^2^ mass on the electrode	Zheng H. et al., 2021 [83]
2-D Bi sheet	BiOCOOH nanowires electrochemically evolved to Bi sheet structure	−0.90, 98%	Catalyst ink drop cast on 0.7 × 0.7 cm^2^ carbon paper (loading 0.25 mg cm^−2^)	Jiang Y. et al., 2023 [85]
Bi_2_O_3_ microfibers	Synthesis by cotton template through simple heating treatment in air.	−0.90, ~100%	Ink painted on L-type glassy carbon	Ning, H., 2023 [84]

**Table 5 materials-17-03077-t005:** Morphologies of bismuth based materials and their band-gaps.

Bismuth-Based Material	Morphology	Bandgap	Reference
Bi_2_WO_6_	Assembly of small wafers	2.88 eV	Li J. et al., 2022 [87]
	Uniform square nanoplates	2.68 eV	Zhou, Y. et al., 2011 [68]
	Nanoflakes	2.75 eV	Li, M. et al., 2015 [69]
BiFeO_3_	Cubic/pseudo-cubic structure(~200 nm)	2.06 eV	Karamian, E. et al., 2018 [88]
BiOCl	Nanosheet;	3.56 eV	Zhang, J. et al., 2021 [89]
	Ultrathin-Nanosheet; ~4 nm thickness	3.38 eV	Zhang, J. et al., 2021 [89]
	Irregularly stacked sheets with flower-like morphology	3.25 eV	Sánchez-Rodríguez, D. et al., 2020 [90]
	Microspheres consisting of nanosheets	3.20 eV	Gao, M. et al., 2019 [91]
	Semi-sphere microparticle with smooth surface	3.60 eV	Hernandez, M. et al., 2023 [92]
BiOBr	Thin nanosheets assembled into a flower like structure	2.60 eV	Gao, M. et al., 2019 [91]
	Nanosheet with diameter of 1.9 to 4.8 µm	2.88 eV	Wu, D. et al., 2017 [42]
	Nanosheet with diameter of 6.8 to 26.4 µm	2.83 eV	Wu, D. et al., 2017 [42]
	Microspherical particles	1.9 eV	Hernandez, M. et al., 2023 [92]

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
