# Peer review of "Role of Facets and Morphologies of Different Bismuth-Based Materials for CO2 Reduction to Fuels"

_materials, 2024, doi:10.3390/ma17133077_

Round 1
Reviewer 1 Report
Comments and Suggestions for Authors
Highlight changes in yellow in a next revision, please. No track changes.
Abstract says very little it needs to start with contextualization methodology and findings and implications.
All abbreviations need to be defined even if now.
The total similarity expressed in the manuscript is significant. It needs to be addressed.
Why repeat table captions from others?
“Table 1. Electrochemical Reduction Potential (E0) Relative to the Standard Hydrogen Elec- trode (SHE) at pH=7 for CO2 reduction reactions.1”
I’m sorry, but you must understand that figures cannot be distraught to just compare the font sizes used in the figures.
“Figure 1:”
Now there’s need to be careful with terms as desired. What does it mean?
“The desired selectivity”
This is just an example of extensive similarity.
“The non-uniform charge distribution between [Bi2O2] and halogen slices due to the strong intralayer covalent bonding of [Bi2O2] slabs and the weak interlayer van der Waals interaction of the halogen double slabs leads to polarization of the related atoms and orbitals that generates the IEF along the crystal orientation perpendicular to the [Bi2O2] and halogen layers. This induced IEF contributes”
Or
“rent and quenched photoluminescence signal of the {001} facet dominant nanosheets as compared to the {010} facet dominant ones [38]. Also, it was found that increasing the exposure percentage of {001} facets of Bi3O4Cl nanosheets induced a stronger IEF resulting in higher photocatalytic activity. It was explained in the work that according to the meas- ured surface voltage and charge density, the IEF magnitude was positively correlated with the percentage of exposed {001} facets (Figure 2 (b)) [38]. Another study by Wang et al. supports the same where they reported the (001) facet to provide more photogenerated charge carriers [39]. However, Li et al. in 2018 worked on how Bi nanoparticles being de- posited on the different facets of BiOBr resulted in alternative charge transfer properties. It was found that Bi nanoparticles being deposited on the (010) facet had better surface charge alteration which is more favorable for interfacial charge separation and transfer as compared to the deposition on (001) facet (Figure 2(c)). A new charge transfer route was established between the contacted Bi nanoparticles and (010) facets of BiOBr along the path of [Bi2O2]2+ → Bi NPs → Br− [40].”
all coming from the SAME source.... Not possible
The figure needs to to start with a caption before the subcaptions. Also, figures are again distorted. And in fact, the letters do not correspond to the ones in the figures...
How is that even possible?
“Figure 2. (a): Internal Electric Field (IEF) direction in BixOyCl with (001) and (010) facets exposed and (b) IEF dependent photocatalytic activity of Bi3O4Cl nanosheets. Reproduced from reference [38] with permission from The Royal Society of Chemistry,2014. (c) Sche- matic illustration of a new charge transfer route established between the contacted Bi NPs and the (010) facet of BiOBr. Reproduced from reference [40] with permission from Else- vier 2018.”
Check others..
Again, because this is supposed to be an original scientific article, I do not understand the inclusion of all these figures, which are not original and in which case the entire caption was copied to. Please also see that these permissions will have to be proved. Even if given they the figures should not be here.
That would be the case of a chapter, not here.
“Figure 5 (a) : Energy levels for molecular orbitals of CO2. (b) Possible coordination 381 modes of CO2 on a metal surface. Reproduced with permission from Elsevier 2020 [61].”
In similar cases, the reference must include authors names.
“Table 3. Morphologies of bismuth based materials and their band-gaps.”
If this is the review where it is the methodology used?
It needs to be clarified in the abstract separate methods section and conclusions too.
Please do not use We...
The conclusion section needs to start with brief contextualization and methodology. Clarify the main findings and practical implications as well as limitations and future prospects. Because this is supposed to be a review, it needs to clarify, which is the originality innovation and novelty in relation to existing reviews and which ones.
Let me be very clear. The similarity found in the text compromises the entire article.
Comments on the Quality of English Languagemoderate
Reviewer 2 Report
Comments and Suggestions for Authors
The review titled "Role of Facets and Morphologies of Different Bismuth-Based Materials for CO2 Reduction to Fuels" attempts to consolidate studies utilizing Bi-based catalysts for photo, electro, and photoelectroreduction to value-added products. Considering the abundance of literature on this topic, it might be more effective to focus on just one application (photo, electro, or photoelectro) given the extensive existing research. Additionally, while the manuscript lacks future insights and valuable suggestions for synthesizing effective Bi-based catalysts, which I believe are crucial for the readers. Consequently, I recommend against accepting the manuscript in its current form.
1. The authors introduce CCS and CCF in the Introduction, but what about CCU?
2. Please provide definitions for abbreviations (e.g., chemical compounds) upon their first appearance in the manuscript.
3. Please adjust the symbol "=" for the various chemical reactions in Table 1. I suggest including data for products that are only discussed within the manuscript.
4. In the Introduction, the authors reference certain manuscripts to underscore the significance of active facets and catalyst morphology. Given the extensive literature on this subject, clarification is needed regarding the criteria for selecting these studies for discussion. What guided the inclusion of specific studies in this discourse?
5. Please incorporate data on the price of Bi (in comparison to other metals), toxicity analysis (versus other metals), and abundance (compared to other metals). Justification for all information should be provided.
6. As this is a review, please include the most pertinent references published to date in the field of photoelectroreduction and electroreduction of CO2 to value-added products using Bi-based catalysts.
7. It would be beneficial to clarify the objective of the review and its novelty compared to existing literature. Please include an explanation.
8. Incorporate a comprehensive table containing information from studies mentioned in subsections such as photocatalysts, electrocatalysts, and photo-electrocatalysts. Additionally, include details about synthesis procedures and electrode deposition methods. This information is valuable for future readers.
9. Divide Section 2.2 and 2.3 into subsections for better organization.
10. The conclusions lack depth. The review should provide valuable insights on synthesizing effective Bi-based catalysts for the reader.
Comments on the Quality of English LanguageSome mistakes and typos have been detected.
Reviewer 3 Report
Comments and Suggestions for Authors
This review paper offers a valuable contribution to the field of CO2 reduction, specifically focusing on the role of facets and morphologies of different bismuth-based materials. A major strength of this review lies in its comprehensive coverage of different bismuth-based materials, including oxides, sulfides, and oxyhalides, as well as their varying morphologies. By discussing the specific role of active facets in each material and how they influence catalytic activity, the authors provide a valuable framework for understanding the relationship between structure and function.The authors have highlighted the key findings and trends in the field.
However, there are a couple of minor issues that could be addressed:
1. The permission for reproduction of Figure 8 should be obtained to avoid any potential copyright issues.
2. The conclusion could benefit from a discussion on how theoretical and simulation studies could contribute to future research in this field.
Reviewer 4 Report
Comments and Suggestions for Authors
This manuscript is titled, “Role of Facets and Mophologies of Different Bismuth-Based Materials for CO2 reduction to Fuels”. Overall, this review article is interesting, but several key issues need to be resolved before it can be published in the “MDPI materials”. This article could be improved by addressing the following issues:
1. The authors should recommend explaining the difference between the photocatalytic and electrochemical CO2 reduction reactions as the article's premise.
2. In Fig. 2b . The authors can check all respective Figures and respective captions.
3. The authors suggest explaining more characterization like FTIR and Raman it is more helpful to understand the surface properties.
4. The authors should explain the evaluation of gas products during the electrochemical CO2 reduction performance.
5. The authors should explain how the catalyst's structural and chemical state changes during the electrolysis.
6. To enhance the readability of the paper and emphasize its relevance in the context of current scientific advancements, it is advisable to incorporate the latest research progress and related literature in the introduction, e.g., doi.org/10.1002/smll.202306165.
7. Minor typos, etc. must be corrected throughout the manuscript.
Comments on the Quality of English LanguageMinor editing of English language required
Round 2
Reviewer 1 Report
Comments and Suggestions for Authors
Highlight changes in yellow in a next revision, please. No track changes.
The overall similarity present in the manuscript is still high.
“Comments 3: The total similarity expressed in the manuscript is significant. It needs to be addressed.
Response 3: We thank the referee for highlighting this point. We made our best efforts to avoid significant similarities along the text of the manuscript. Moreover, we checked the manuscript for plagiarisms using the Turnitin software provided by our University. Excluding the Reference section (which obviously reports the same words already published in the literature), we obtained an acceptable level of originality in our manuscript”
See that beyond this, the criteria needs to be referenced and part of this caption should, in fact, be in the text, not here.
“Figure 1. Scopus search results providing the number of articles published until May, 2024 with the keywords “Element name”+ ”Photocatalytic/Electrocatalytic/Photoelectrochemical”+ “CO2” + “reduction””
There is no coherent in terms of how the values and units are present, considering the heading.
“Abundance in Earth’s crust (mg/kg)”
Table 2
This is just an example of how similarity should be handled. Please see that in this case, we do not even have a reference.
“The non-uniform charge distribution between [Bi2O2] and halogen slices due to the strong intralayer covalent bonding of [Bi2O2] slabs and the weak interlayer van der Waals interaction of the halogen double slabs lead to polarization of the related atoms and orbitals. I”
Or
“) faceted catalyst indicated that the interplanar spacings of 0.165 nm and 0.240 nm corresponded to (332) and (202) facets of tetragonal phase CuBi2O4 and the interplanar spacings of 0.253, 0.243 and 0.407 nm were assigned to (”
or
“behavior was mainly attributed to the formation of a Schottky junction on the reduction facet of the semiconductor. Anchoring a metal on that facet enhanced hot-electron injec- tion into the metal. This is due to the high density of hot electrons on the reduction facet and the enhanced surface electric states on the semiconductor due to metal modification, which ultimately accelerates electron transfer to metal. There is an increase in the efficient charge separation due to the unidirectional electron transfer route from the semiconduc- tor to the metal. CO2 photoreduction yielded CH4 and CO with generation rates being 2.6 and 1.8 times respectively higher for BiVO4{010}-Au-Cu2O than that for BiVO4{110}-Au- Cu2O[56].”
or
“like Bi can convert the C of CO2 to HCOOH through protonation due to high hydrogen evolution overpotential and weak adsorption of CO2• Ì…intermediate.”
Or
“d 42.1% of the absorbed photon-to-current conversion efficiency at 1.23V vs RHE. Mainly, the solar light conversion efficiency is directly proportional to the product of the solar light absorption efficiency, charge separation efficiency, and surface charge transfer efficiency. Therefore, the enhanced PEC performance of the BiVO4 nano- plates was due to the interfacial electron transport reaction between the {010} plane and the electrolyte with b”
only to illustrate my point.
And it is worrying when we start seeing content from the same source and, in fact, references cited differ.
Similarity is a red sign of alert globally. It means the authors either entirely copy paste to content or are not aware of the need to rewrite using their own words.
In addition, the similarity mentioned in this particular subsection, for example, comes from the same source every time.
“2.1. Photocatalysis”
Check similarity.
“Figure 2. Representation of different charge transfer routes (a): Internal Electric Field (IEF) direction in BixOyCl with (001) and (010) facets exposed and (b) IEF dependent photocata- lytic activity of Bi3O4Cl nanosheets. Reproduced from reference [46] with permission from The Royal Society of Chemistry,2014. (c) Schematic illustration of a new charge transfer route established between the contacted Bi NPs and the (010) facet of BiOBr. Reproduced from reference [48] with permission from Elsevier 2018.”
I really do not understand how it is ....similarity in the caption of a figure
And again, if the figures are published, why mention them here again in a supposedly original article, just remove them the same in every other case.
“Figure 4 : SEM images of (a) Bi2S3 microspheres (b) Bi2S3 microspheres with packed nano- plates. Reproduced from reference [66] with permission from The Royal Society of Chem- istry,2019.”
Please see that such expressions are not assertive nor clear, nor can they be considered scientific.
“as has been discussed so far,”
“to the inundated modern human needs”
Going back and because authors clearly identify these manuscript as a review, I do not find a clear methodology being presented. Which database is what time frame which criteria?
Besides a separate section this methodology needs to be clear also in the abstract and in the conclusion section In addition, the authors need to highlight the novelty of this review compared to others. Where is the innovation?
why scopus?
Comments on the Quality of English Languagemoderate
Reviewer 2 Report
Comments and Suggestions for Authors
I recommend its publication in its current form.
Author Response
We thanks the reviewer for appreciating our work.
Reviewer 4 Report
Comments and Suggestions for Authors
The authors have well addressed all my comments. Therefore, present form of the revised manuscript is ready for publication.
Author Response

(The authors gave the same response as above.)
